# Co-Inoculations with Plant Growth-Promoting Bacteria in the Common Bean to Increase Efficiency of NPK Fertilization

Emariane Satin Mortinho [1], Arshad Jalal [1], Carlos Eduardo da Silva Oliveira [1], Guilherme Carlos Fernandes [1], Nathália Cristina Marchiori Pereira [1], Poliana Aparecida Leonel Rosa [1], Vagner do Nascimento [2], Marco Eustáquio de Sá [1] and Marcelo Carvalho Minhoto Teixeira Filho [1,*]

[1] School of Engineering (FEIS), São Paulo State University (UNESP), Ilha Solteira 15385-000, SP, Brazil; emariane.satin@unesp.br (E.S.M.); arshad.jalal@unesp.br (A.J.); ces.oliveira@unesp.br (C.E.d.S.O.); guilherme.carlos.fernandes@gmail.com (G.C.F.); marchiori.nathalia@gmail.com (N.C.M.P.); polianaleonelrosa@gmail.com (P.A.L.R.); marcosa@agr.feis.unesp.br (M.E.d.S.)

[2] College of Agricultural and Technological Sciences (FCAT), São Paulo State University (UNESP), Dracena 17900-000, SP, Brazil; vagner.nascimento@unesp.br

\* Correspondence: mcm.teixeira-filho@unesp.br

**Abstract:** Given the hypothesis that co-inoculation with plant growth-promoting bacteria (PGPB) enhances the beneficial effects of *Rhizobium tropici* with greater mineral nutrition, optimizes biological nitrogen fixation and reduces use of fertilizers in bean plants, the objective of this research was to evaluate the synergistic effects of *Rhizobium tropici* associated with *Azospirillum brasilense*, *Bacillus subtilis*, *Pseudomonas fluorescens* and their combinations, on increasing the efficiency of NPK fertilization to obtain high winter yields of the (irrigated) common bean in the Cerrado region. The experiment was carried out in the field over two years in a Rhodic Hapludox under a no-till system in Selvíria, Brazil. The experimental design comprised complete randomized blocks with four replications in a $3 \times 7$ factorial scheme. The treatments consisted of three doses of NPK fertilizer (control—0 kg ha$^{-1}$ (control); 50% of the recommended dose; 100% of the recommended dose in two parts) and seven doses of inoculation or co-inoculation (control; *Rhizobium tropici*; *R. tropici* + *Azospirillum brasilense*; *R. tropici* + *Bacillus subtilis*; *R. tropici* + *Pseudomonas fluorescens*; *R. tropici* + *A. brasilense* + *B. subtilis*; *R. tropici* + *A. brasilense* + *P. fluorescens*). The PGPB in the co-inoculations increased the hundred-grain weight, the grain pod$^{-1}$, the grain plant$^{-1}$ and the grain yield following the NPK doses. The grain yield of the common bean was increased by co-inoculation with *R. tropici* + *A. brasilense* + *P. fluorescens* without NPK treatments, co-inoculation with *R. tropici* + *P. fluorescens* and *R. tropici* + *A. brasilense* + *B. subtilis* with the 50% dose of NPK and co-inoculation with *R. tropici* + *B. subtilis* with the recommended dose of NPK fertilizer (100%).

**Keywords:** *Phaseolus vulgaris* L.; grain yield; fertilizer reduction; bacteria–fertilizer integration; co-inoculation; *Azospirillum brasilense*; *Bacillus subtilis*; *Pseudomonas fluorescens*



## 1. Introduction

Brazil is one of the main producers of the common bean (*Phaseolus vulgaris* L.) and is ranked third in the world, with 3200 tons being produced in a cultivated area of 2900 hectares [1]. It is the most consumed legume in the human diet in Brazil due to its richness in proteins, oils, vitamins and amino acids [2]. The common bean is a plant with a high demand for fertilizer; therefore, cultivation in acidic soils with low fertility could affect grain yields due to the loss of nitrogen (N) and potassium (K) fertilizers by leaching and volatilization and the loss of phosphate (P) by adsorption as a result of binding with iron (Fe) and aluminum (Al) oxides in soil [3].

The soils of the Brazilian region of Cerrado are highly weathered and acidic, with low fertility and high concentrations of Fe and Al. Thus, increased P adsorption in the soils reduces availability for plant absorption; therefore, high amounts of fertilizers and

correctives are applied to achieve greater yields [4]. Bean cultivation is characterized by an intensive use of synthetic fertilizers to increase productivity; however, it can also increase the cost of production and cause serious consequences to the agroecosystem [5]. Hence, the introduction of plant growth-promoting bacteria (PGPB) to crop systems is one of the most sustainable and economically affordable strategies to improve soil quality, as well as discourage the use of chemical fertilizers and pesticides [6].

Plant growth-promoting bacteria adopt multiple mechanisms, such as the production and secretion of phytohormones (indole-3-acetic acid (IAA), cytokinins, gibberellins and ethylene) and plant growth regulators (abscisic acid, nitric oxide and polyamines), to promote plant growth, increase nutrient availability and phosphate solubilization and protect plants against biotic and abiotic stresses [7]. These PGPB promote biological nitrogen fixation (BNF) to increases the activity of nitrate reductase and the efficiency of N use in plants [8].

The concept of co-inoculation is being studied using combinations of different microorganisms to increase the availability and acquisition of nutrients, early nodulation and yields [9]. The co-inoculation of common bean seeds with *Rhizobium tropici* and *Azospirillum brasilense* is commonly practiced in Brazil to stimulate plant nodulation and increase grain yield [10]. In addition, co-inoculation with *Rhizobium* sp. and *Bacillus* sp. stimulates symbioses that favor the process of BNF and phosphate solubilization [11], while *Pseudomonas* sp. and *R. tropici* promote plant growth and productivity, along with a greater acquisition of N by the bean plant [12]. Inoculation with these bacteria has the ability to reduce the consumption of synthetic fertilizers, such as nitrogen [8,9], phosphate [11,13,14] and NPK, at seeding [6]. Considering the lack of information regarding the effects of co-inoculation on reducing the use of synthetic fertilizers for the common bean, it was hypothesized that co-inoculation could improve plant nutrition, optimize biological nitrogen fixation and reduce the use of fertilizers for the common bean. In this context, the objective of this research was to evaluate the synergistic effects of *Rhizobium tropici*, plus *R. tropici* when associated with *Azospirillum brasilense*, *Bacillus subtilis*, *Pseudomonas fluorescens* and their combinations, on increasing the efficiency of NPK fertilization to obtain high winter yields of the (irrigated) common bean in the Brazilian region of Cerrado.

## 2. Materials and Methods

### 2.1. Location and Characterization of Experimental Area

The experiment was conducted in 2019 and 2020 at the Teaching, Research and Extension Farm of the Faculty of Engineering, São Paulo State University (UNESP), Ilha Solteira Campus, São Paulo, Brazil. The site is located in the municipality of Selvíria, State of Mato Grosso do Sul ($51°22'$ W longitude and $20°22'$ S latitude), with an altitude of 335 m.

The region has a humid tropical climate, with a rainy season in the summer and a dry season in the winter and an average rainfall of 1322 mm and an average annual temperature of 23 °C, which is classified as Aw type [15]. The soil of the experimental site is classified as a Rhodic Hapludox with a clay-like texture [16]. A composite soil sample was collected from 0.00 to 0.20 m layers before the installation of the experiment, which was analyzed according to methodology proposed by Raij et al. [17] for chemical attributes: P $_{(resin)}$ = 38 mg dm$^{-3}$; S-SO$_4$ = 5 mg dm$^{-3}$; organic matter = 23 g dm$^{-3}$; pH (CaCl$_2$) = 5.2; K, Ca, Mg and H + Al = 3.0, 39.0, 33.0 and 34.0 mmol$_c$ dm$^{-3}$, respectively; Cu, Fe, Mn and Zn $_{(DTPA)}$ = 6.1, 29.0, 107.7 and 1.4 mg dm$^{-3}$, respectively; B $_{(hot\ water)}$ = 0.15 mg dm$^{-3}$ with 69% base saturation. Climatic data referring to the periods of experimentation during the winter seasons of 2019 and 2020 are summarized in Figure 1.

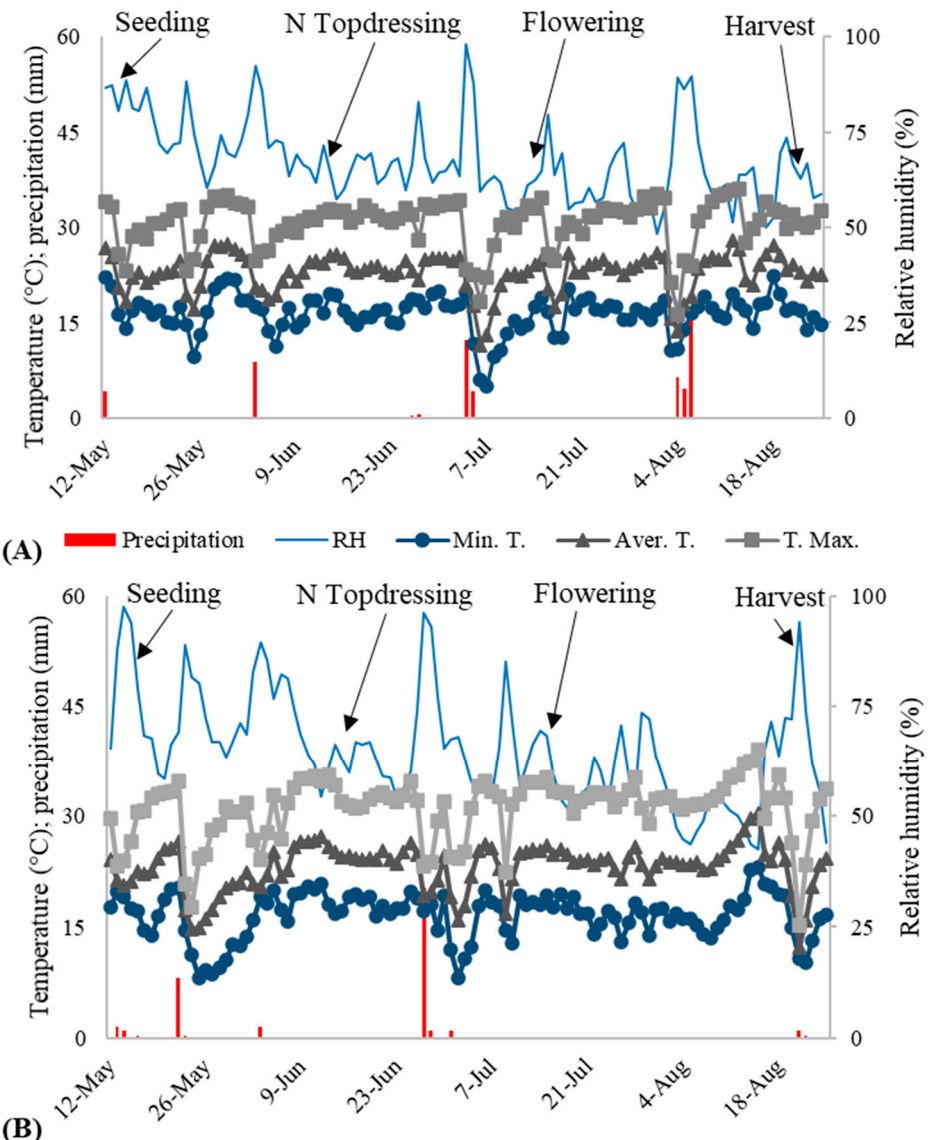

**Figure 1.** Maximum, average and minimum temperatures and precipitation and the relative humidity during the winter growth periods of the common bean in 2019 (**A**) and 2020 (**B**).

### 2.2. Experimental Design

The experiment was carried out in randomized blocks with four replications in a $7 \times 3$ factorial scheme. The treatments were composed of inoculations and co-inoculations (1 = un-inoculated-control; 2 = inoculation with *R. tropici*; 3 = co-inoculation with *R. tropici* + *A. brasilense*; 4 = co-inoculation with *R. tropici* + *B. subtilis*; 5 = co-inoculation with *R. tropici* + *P. fluorescens*; 6 = co-inoculation with *R. tropici* + *A. brasilense* + *B. subtilis*; 7 = co-inoculation with *R. tropici* + *A. brasilense* + *P. fluorescens*) and reduced doses of NPK during seeding and coverage fertilization (unfertilized; 50% of the recommended dose; 100% of the recommended dose in two parts).

### 2.3. Installation of Field Research

The experimental site had been under a no-till system for 16 years and was sown with soybean prior to common bean cultivation in both years. A winter cultivar of the common bean IPR Campos Gerais of the carioca type from the IAPAR group was selected for the experiment, which has a type II indeterminate growth habit and an average yield potential of 3987 kg ha$^{-1}$.

The seeds were treated with STANDAK$^{®}$ TOP (a co-formulation of pyraclostrobin, thiophanate methyl and fipronil-active ingredients (a.i.)), based on the manufacturer's recommendations for the crop. The seeds were then inoculated with *Rhizobium tropici* (SEMIA 4077) at a dose of 2 g per kg$^{-1}$ of seeds (guaranteed $2 \times 10^9$ colony forming units, CFU g$^{-1}$). A 10% sugar solution was added during inoculation to facilitate inoculant adhesion to the seeds.

The co-inoculations were performed according to the recommendation of the company that provided the inoculants, Total Biotecnologia$^{™}$. Inoculations with the *Azospirillum brasilense* strains (Ab-V5 and Ab-V6) at a dose of 100 mL of liquid inoculant (guaranteed $2 \times 10^8$ CFU mL$^{-1}$), the *Bacillus subtilis* strain (CCTB04) at a dose of 50 mL (guaranteed $1 \times 10^8$ CFU mL$^{-1}$) and the *Pseudomonas fluorescens* strain (CCTB03) at a dose of 100 mL of liquid inoculant (guaranteed $2 \times 10^8$ CFU mL$^{-1}$) per 50 kg seeds were performed. Seed inoculations and co-inoculations were carried out an hour before the common bean sowing.

The common bean sowing was carried out mechanically on 15 May 2019 and 16 May 2020 using a furrower rod mechanism, with 16 seeds per meter and a spacing of 0.45 m between the rows. Each plot consisted of four lines of 5 m, which were spaced by 0.45 m. Based on the soil analysis, crop history in the area and the recommendations of Ambrosano et al. [17], the common bean was applied with recommended dose (100%) of NPK at the time of sowing: 40 kg ha$^{-1}$ N from urea, 80 kg ha$^{-1}$ P$_2$O$_5$ from single superphosphate and 40 kg ha$^{-1}$ K$_2$O from potassium chloride. Following the limit interpretation of the Instituto Agronômico de Campinas (IAC), the soil of experiment site was low in boron (B) content; therefore, all treatments were applied with foliar spray at a dose of 1 kg ha$^{-1}$ of B from boric acid during the pre-flowering stage in both years.

Cover fertilization with N (90 kg ha$^{-1}$ from urea) was carried out between the lines of common bean plants 30 days after the emergence of the plants. The area was irrigated by a central pivot, with an average depth of 14 mm, every three days or whenever necessary.

### 2.4. Assessments

The trifoliate leaves were randomly collected during the full flowering stage of the common bean in July 2019 and 2020. The collected leaves were analyzed for concentrations of N, P and K from nitric–perchloric digestion, according to the methodology of Malavolta et al. [18]. Five plants were collected from the useful area of each plot at the time of harvest for the evaluation of production components (number of pods per plant$^{-1}$, number of grains per pod, number of grains per plant and hundred-grain weight). Grain yields were determined by manual collection from the plants in the two central lines of each plot. The plants were exposed to the sun for drying and were then threshed by a mechanical thresher to calculate the grain weight, which was transformed into kg ha$^{-1}$ at 13% (wet basis).

### 2.5. Statistical Analysis

The obtained data were submitted to an analysis of variance using an F-test ($p < 0.05$). The Scott–Knott test at a 5% probability was used to compare the means of the co-inoculations while Tukey's test at a 5% probability was used to compare the means of the treatments with NPK fertilizer. A Pearson's correlation analysis was also performed on the parameters that were evaluated in the treatments using an SAS program.

## 3. Results

### 3.1. Effects of Fertilizer Dose Reduction with Inoculation on Leaf N, P and K Concentrations

There was a significant ($p < 0.01$) effect of NPK doses, inoculations and their interactions on leaf N concentration in 2019, leaf P concentration in 2020 and leaf K concentration in both 2019 and 2020, while their effects were not significant ($p > 0.05$) for leaf N concentration in 2020 or leaf P concentration in 2019 (Table 1). Higher leaf N concentrations in 2019 were observed with the 100% dose of NPK in relation to the reduced doses. In addition, higher leaf P and K concentrations in both years were observed with reduced NPK doses

(0 and 50% doses) compared to the 100% dose. The co-inoculation of the common bean with *R. tropici* + *A. brasilense* was observed to produce a higher leaf N concentration in 2019, while inoculation with *R. tropici* and co-inoculation with *R. tropici* + *B. subtilis*, *R. tropici* + *A. brasilense* + *B. subtilis* and *R. tropici* + *A. brasilense* + *P. fluorescens* were observed to produce higher leaf P concentrations in 2020 compared to the other inoculations. Co-inoculation with *R. tropici* + *A. brasilense* + *P. fluorescens* and non-inoculation were noted as producing the highest leaf K concentration in 2019 whereas leaf K concentration in 2020 was higher following treatments of no inoculation, inoculation with *R. tropici* and co-inoculations with *R. tropici* + *B. subtilis*, *R. tropici* + *P. fluorescens* and *R. tropici* + *A. brasilense* + *P. fluorescens* (Table 1).

**Table 1.** Leaf concentrations of nitrogen, phosphorus and potassium in relation to NPK doses and co-inoculations with PGPB in common bean seeds in 2019 and 2020.

| NPK Doses (%) | Leaf N Concentration (g kg$^{-1}$) | | Leaf P Concentration (g kg$^{-1}$) | | Leaf K Concentration (g kg$^{-1}$) | |
|---|---|---|---|---|---|---|
| | 2019 | 2020 | 2019 | 2020 | 2019 | 2020 |
| 0 | 50.10 b | 45.51 | 6.29 | 3.89 a | 28.68 a | 22.14 a |
| 50 | 48.74 c | 44.16 | 6.34 | 3.74 ab | 27.38 a | 21.27 a |
| 100 | 51.99 a | 44.23 | 6.21 | 3.60 b | 24.85 b | 19.94 b |
| **LSD (%)** | 1.08 | 1.63 | 0.32 | 0.20 | 1.54 | 1.13 |
| **Inoculation** | | | | | | |
| Non-Inoculation | 50.59 b | 44.38 | 6.42 | 3.57 b | 30.33 a | 22.69 a |
| *R. tropici* | 49.99 b | 43.56 | 6.20 | 3.81 a | 27.10 b | 22.02 a |
| *R. tropici* + *A. brasilense* | 52.31 a | 43.38 | 6.37 | 3.47 b | 26.98 b | 19.66 b |
| *R. tropici* + *B. subtilis* | 47.33 c | 45.67 | 5.99 | 3.94 a | 24.81 c | 20.93 a |
| *R. tropici* + *P. fluorescens* | 49.80 b | 45.47 | 6.33 | 3.57 b | 26.48 b | 21.32 a |
| *R. tropici* + *A. brasilense* + *B. subtilis* | 51.12 b | 45.19 | 6.10 | 3.82 a | 22.92 c | 19.89 b |
| *R. tropici* + *A. brasilense* + *P. fluorescens* | 50.79 b | 44.80 | 6.56 | 4.02 a | 30.17 a | 21.30 a |
| **F-Test** | | | | | | |
| NPK Doses (D) | 27.14 ** | 2.57 ns | 0.51 ns | 6.34 ** | 19.02 ** | 11.49 ** |
| Co-Inoculation (C) | 10.37 ** | 1.55 ns | 1.88 ns | 5.58 ** | 15.31 ** | 4.71 ** |
| D × C | 6.96 ** | 1.97 ns | 0.99 ns | 2.74 ** | 7.44 ** | 3.36 ** |
| General Mean | 50.28 | 44.63 | 6.82 | 3.74 | 26.97 | 21.12 |
| Standard Error | 0.31 | 0.47 | 0.09 | 0.06 | 0.44 | 0.33 |
| **CV (%)** | 2.85 | 4.88 | 6.82 | 7.17 | 7.59 | 7.09 |

Means followed by same letter within a column did not differ statistically from each other, according to the Scott–Knott test, neither did those within a row, according to Tukey's test at a 5% probability: ** significant at $p < 0.01$; ns not significant.

The interactions between the NPK doses and the inoculations were significant ($p > 0.05$) for leaf N concentration in 2019 (Figure 2A). Higher leaf N concentrations were observed with 100% doses of NPK fertilizer after no inoculation and co-inoculation with *R. tropici* + *A. brasilense* + *P. fluorescens*. There were no differences in leaf N concentration following inoculation with *R. tropici* and *R. tropici* + *A. brasilense*, regardless of the dose of NPK fertilizer. The treatments with 50% dose of NPK and inoculation with *R. tropici* and *R. tropici* + *A. brasilense* resulted in the highest leaf N concentrations. In addition, co-inoculations with *R. tropici* + *A. brasilense* and *R. tropici* + *P fluorescens* were observed to produce higher leaf N concentrations with the 0% dose of NPK fertilizer.

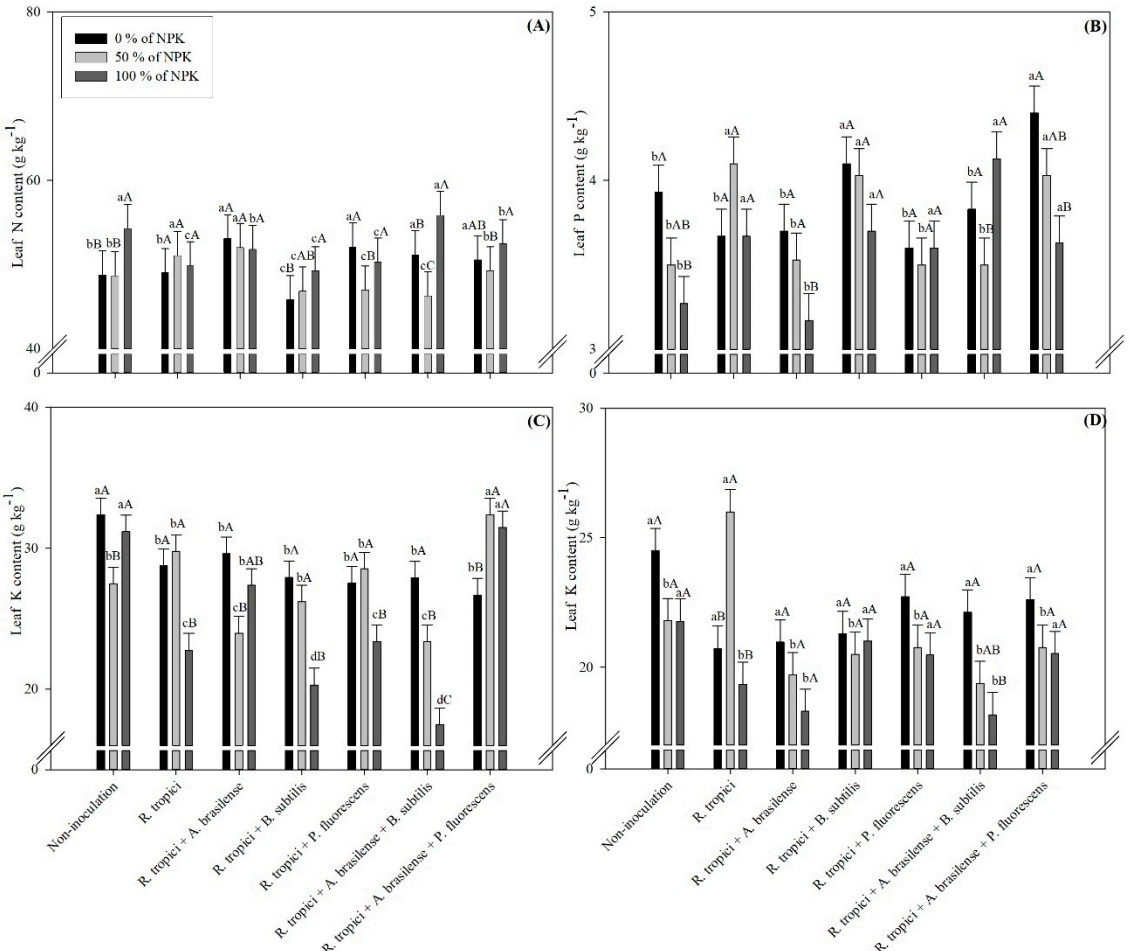

**Figure 2.** Effects of interactions between reduced doses of NPK fertilizer and co-inoculation with PGPB in common bean seeds on leaf N concentration in 2019 (**A**), leaf P concentration in 2020 (**B**) and leaf K concentration in 2019 (**C**) and 2020 (**D**). Means followed by the same capital letter differ from each other (different colors refer to the different doses of NPK fertilizer with each inoculation), according to Tukey's test at a 5% probability. Means followed by different lowercase letters differ from each other (bars of the same color refer to each inoculation with different doses of NPK fertilizer), according to the Scott–Knott test at a 5% probability.

There was a significant interaction between NPK dose and inoculations for leaf P concentrations in 2020 (Figure 2B). Higher leaf P concentrations were observed with the 0% dose of NPK fertilizer and co-inoculation with *R. tropici + A. brasilense + P. fluorescens* and *R. tropici + B. subtilis*, while a 50% dose of NPK fertilizer along with co-inoculation with *R. tropici* increased leaf P concentration, which was statistically similar to *R. tropici + B. subtilis* co-inoculation. The application of the 100% dose of NPK along with co-inoculation with *R. tropici + A. brasilense + B. subtilis* also increased leaf P concentration in the common bean, which was statistically not different from the treatments inoculated with *R. tropici*, *R. tropici + B. subtilis* and *R. tropici + P fluorescens*. In addition, the lowest leaf P concentration was observed with the 100% dose of NPK fertilizer and co-inoculation with *R. tropici + A. brasilense* (Figure 2B).

The interactions between leaf K concentrations were verified as being significant in the common bean cultivation in both 2019 and 2020 (Figure 2C,D). The highest leaf K concentration in first year was observed following the application of the 50% dose of NPK fertilizer and *R. tropici + A. brasilense + P. fluorescens* treatment, which was statistically similar to the 0 and 100% doses of NPK with non-inoculation treatments and the 100% dose of NPK with co-inoculation with *R. tropici + A. brasilense + P. fluorescens*. In addition, a 50% dose of

NPK and inoculation with *R. tropici* provided higher leaf K concentrations in second season of common bean cultivation (Figure 2D). There were no significant differences between leaf K concentrations in 2020 following the 0 and 100% doses of NPK with treatments of no inoculation and co-inoculation with *R. tropici + A. brasilense*, *R. tropici + B. subtilis*, *R. tropici + P. fluorescens* and *R. tropici + A. brasilense + P. fluorescens*. The lowest leaf K concentrations in both seasons were observed for the 100% dose of NPK and co-inoculation with *R. tropici + A. brasilense + B. subtilis* (Figure 2C,D).

### 3.2. Effects of Fertilizer Dose Reduction with Inoculation on Yield Components

There was a significant effect ($p > 0.01$) of NPK dose on the hundred-grain weight in 2019 and 2020, the number of pods per plant in 2020 and the number of grains per plant in 2019 and 2020, while inoculation was significant ($p > 0.01$) for the hundred-grain weight in 2020, the number of grains per pod in 2020 and the number of grains per plant in 2019 (Table 2). The highest hundred-grain weight was observed following the 100 and 50% doses of NPK fertilizer in 2019 and 2020. The highest numbers of pods per plant and grains per plant were observed following the 50 and 100% doses of NPK in both cropping seasons. In addition, co-inoculation with *R. tropici + B. subtilis* provided greater hundred-grain weights in 2019, while non-inoculation provided a greater number of grains per pod in 2020 compared to the other inoculations. The number of grains per plant increased following co-inoculation with *R. tropici + B. subtilis*, which was statistically similar to co-inoculation with *R tropici + A. brasilense*, *R. tropici + P. fluorescens* and *R. tropici + A. brasilense + P. fluorescens* (Table 2).

**Table 2.** Hundred-grain weight, number of pods per plant, number of grains per pod and number of grains per plant in relation to NPK doses and co-inoculation with PGPB in common bean seeds in 2019 and 2020.

| NPK Doses (%) | 100-Grain Weight (g) | | Pods per Plant | | Grains per Pod | | Grains per Plant | |
|---|---|---|---|---|---|---|---|---|
| | 2019 | 2020 | 2019 | 2020 | 2019 | 2020 | 2019 | 2020 |
| 0 | 25.26 c | 25.61 b | 23.79 | 20.61 b | 4.74 | 4.96 | 115.6 a | 101.5 b |
| 50 | 25.87 b | 26.47 a | 25.36 | 26.50 a | 4.69 | 4.85 | 125.2 a | 125.3 a |
| 100 | 26.43 a | 25.88 b | 23.18 | 24.89 a | 4.73 | 5.22 | 100.2 b | 128.8 a |
| **LSD (%)** | 0.27 | 0.50 | 4.06 | 3.18 | 0.06 | 0.69 | 10.61 | 20.23 |
| **Inoculation** | | | | | | | | |
| Non-Inoculation | 25.72 c | 26.00 | 23.25 | 20.83 | 4.80 | 6.65 a | 106.8 b | 136.2 |
| *R. tropici* | 26.01 b | 26.51 | 22.75 | 25.50 | 4.70 | 4.79 b | 93.9 b | 121.3 |
| *R. tropici + A. brasilense* | 25.95 c | 25.93 | 23.83 | 24.08 | 4.79 | 5.41 b | 121.2 a | 127.7 |
| *R. tropici+ B. subtilis* | 26.40 a | 25.35 | 26.17 | 25.75 | 4.65 | 4.75 b | 128.0 a | 120.4 |
| *R. tropici + P. fluorescens* | 25.99 b | 25.88 | 25.17 | 24.25 | 4.70 | 4.49 b | 120.9 a | 108.5 |
| *R. tropici + A. brasilense + B. subtilis* | 26.10 b | 26.19 | 23.17 | 24.92 | 4.74 | 4.56 b | 103.0 b | 114.0 |
| *R. tropici + A. brasilense+ P. fluorescens* | 25.80 c | 26.06 | 24.42 | 22.67 | 4.65 | 4.42 b | 121.7 a | 101.5 |
| **F-Test** | | | | | | | | |
| NPK Doses (D) | 53.97 ** | 8.97 ** | 0.89 ns | 10.58 ** | 0.19 ns | 0.85 ns | 16.43 ** | 6.23 ** |
| Co-Inoculation (C) | 12.60 ** | 2.48 ns | 0.45 ns | 1.46 ns | 0.42 ns | 6.48 ** | 6.82 ** | 1.65 ns |
| D × C | 13.97 ** | 1.35 ns | 1.51 ns | 1.20 ns | 5.06 ** | 1.54 ns | 6.77 ** | 1.46 ns |
| General Mean | 25.85 | 25.99 | 24.11 | 24.00 | 4.72 | 5.01 | 113.35 | 118.55 |
| Standard Error | 0.08 | 1.15 | 1.19 | 0.94 | 0.06 | 0.20 | 3.12 | 5.95 |
| **CV (%)** | 1.63 | 2.98 | 26.21 | 20.65 | 7.05 | 21.55 | 14.52 | 26.56 |

Means followed by same letter within a column did not differ statistically from each other, according to the Scott–Knott test, neither did those within a row, according to Tukey's test at a 5% probability: ** significant at $p < 0.01$; ns not significant.

The interactions between NPK fertilizer dose and the inoculations were significant ($p > 0.01$) for hundred-grain weight in 2019 (Figure 3A). Greater hundred-grain weights were noted following the 100% dose of NPK fertilizer and co-inoculation with *R. tropici +*

*A. brasilense*, which was statistically similar to the 50% dose of NPK and co-inoculation with *R tropici* + *P. fluorescens*, as well as the 0% dose of NPK and co-inoculation with *R. tropici* + *B. subtilis*. Inoculation with *R. tropici* and the 0% dose of NPK fertilizer was observed to produce the lowest hundred-grain weights, compared to the control and co-inoculation combinations (Figure 3A).

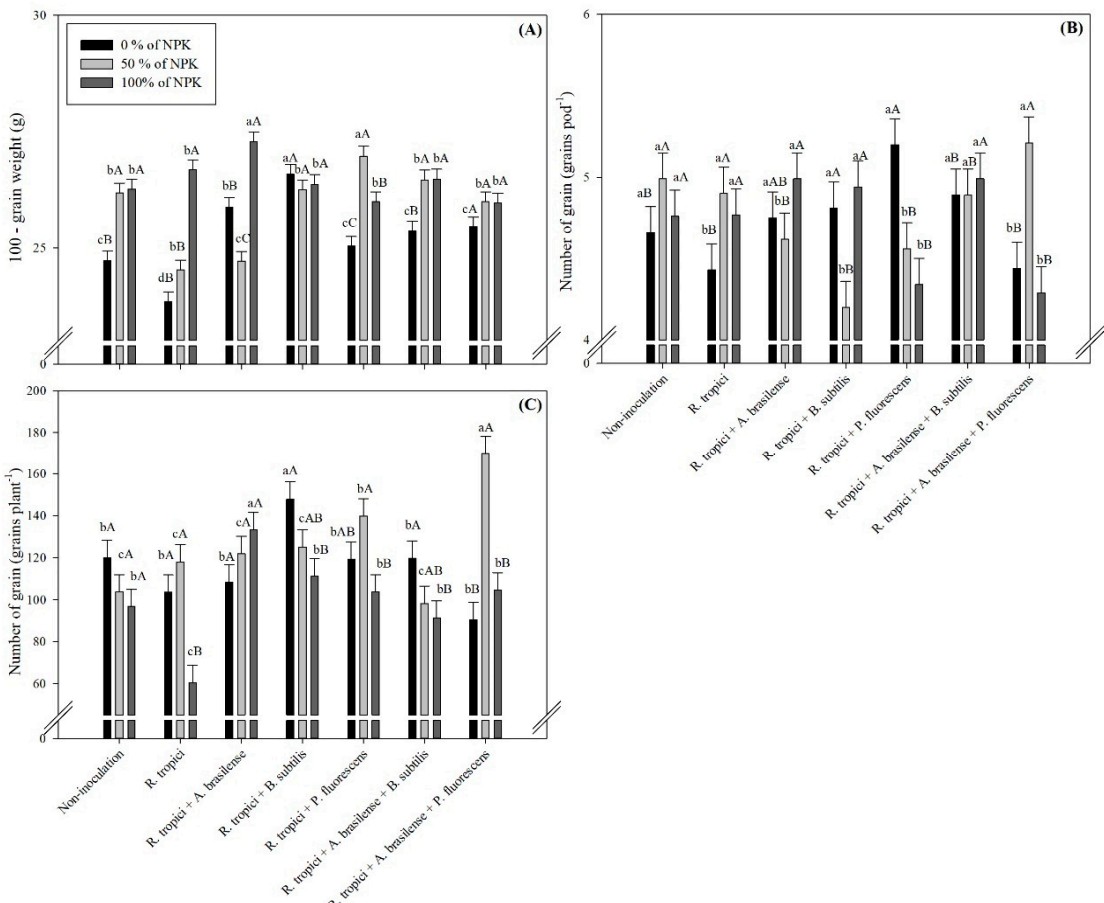

**Figure 3.** Effects of interactions between reduced doses of NPK fertilizer and co-inoculation with PGPB in common bean seeds on hundred-grain weight in 2019 (**A**), the number of grains per pod in 2019 (**B**) and the number of grains per plant in 2019 (**C**). Means followed by the same capital letter differ from each other (different colors refer to the different doses of NPK fertilizer with each inoculation), according to Tukey's test at a 5% probability. Means followed by different lowercase letters differ from each other (bars of the same color refer to inoculation with different doses of NPK fertilizer), according to the Scott–Knott test at a 5% probability.

The interactions between the inoculations and NPK fertilizer doses were significant on the number of grains per pod during the first harvest of the common bean (Figure 3B). The highest number of grains per pod was noted following the 50% dose of NPK and co-inoculation with *R. tropici* + *A. brasilense* + *P. fluorescens*, which was statistically not different from the 50% dose of NPK with no inoculation and inoculation with *R. tropici* and the 100% dose of NPK and co-inoculation with *R. tropici* + *A. brasilense* + *B. subtilis*, *R. tropici* + *A. brasilense*, *R. tropici* + *B. subtilis*, *R. tropici* and no inoculation, as well as the 0% dose of NPK and co-inoculation with *R. tropici* + *P. fluorescens*, *R. tropici* + *A. brasilense* and *R. tropici* + *B. subtilis*. The lowest number of grains per pod were observed following the 50% dose of NPK and co-inoculation with *R. tropici* + *B. subtilis*, in relation to the 0 and 100% doses of NPK and the other inoculations (Figure 3B).

The highest number of grains per plant were noted following the 50% dose of NPK fertilizer and co-inoculation with *R. tropici* + *A. brasilense* + *P. fluorescens*, which was statisti-

cally not different from the 0% dose of NPK fertilizer and co-inoculation with *R. tropici +
B. subtilis* (Figure 3C). The lowest number of grains per plant was observed following the
100% dose of NPK and inoculation with *R. tropici*, compared to no inoculation and all other
co-inoculations and the 0 and 50% doses of NPK fertilizer (Figure 3C).

### 3.3. Effects of Fertilizer Dose Reduction with Inoculation on Grain Yield

There was a significant ($p < 0.01$) effect of NPK dose, inoculation and their interactions
on grain yield in 2019 and 2020 (Table 3). The application of the 50% dose of NPK fertilizer
was observed to produce a greater grain yield of common bean in both studied years,
compared to the 0 and 100% doses of fertilizer. In addition, co-inoculation with *R. tropici
+ B. subtilis* in 2019 and co-inoculation with *R. tropici + A. brasilense + P. fluorescens* in
2020 produced greater grain yields compared to no inoculation and all other inoculations
(Table 3).

**Table 3.** Means of grain yield in relation to doses of NPK and co-inoculations with PGPB in common
bean seeds in 2019 and 2020.

| NPK Doses (%) | Grain Yield (kg ha$^{-1}$) | |
| --- | --- | --- |
| | **2019** | **2020** |
| 0 | 3212 b | 2751 c |
| 50 | 3693 a | 3813 b |
| 100 | 3822 a | 4370 a |
| **LSD (%)** | 246.52 | 248.70 |
| **Inoculation** | | |
| Non-Inoculation | 3527 b | 3435 b |
| *R. tropici* | 3203 c | 3464 b |
| *R. tropici + A. brasilense* | 3461 b | 3534 b |
| *R. tropici + B. subtilis* | 4329 a | 3571 b |
| *R. tropici + P. fluorescens* | 3240 c | 3712 b |
| *R. tropici + A. brasilense + B. subtilis* | 3665 b | 3664 b |
| *R. tropici + A. brasilense + P. fluorescens* | 3605 b | 4131 a |
| **F-Test** | | |
| NPK Doses (D) | 19.67 ** | 126.44 ** |
| Co-Inoculation (C) | 11.44 ** | 4.47 ** |
| D × C | 9.39 ** | 2.67 ** |
| General means | 3575 | 3644 |
| Standard error | 72.51 | 76.16 |
| **CV (%)** | 10.73 | 10.63 |

Means followed by same letter within a column did not differ statistically from each other, according to the
Scott–Knott test, neither did those within a row, according to Tukey's test at a 5% probability: ** significant at
$p < 0.01$.

The greatest grain yield from the first harvest was noted following the 100% dose
of NPK and co-inoculation with *R. tropici + B. subtilis*, which was statistically similar to
treatments applied with the 50% dose of NPK fertilizer and co-inoculation with *R. tropici +
A. brasilense, R. tropici + P. fluorescens* and *R. tropici + A. brasilense + B. subtilis*, in relation
to the 0% dose of NPK fertilizer and the other inoculations (Figure 4A). The lowest grain
yield from this harvest was noted following the 100% dose of NPK and co-inoculation with
*R. tropici + P. fluorescens* (Figure 4A). In 2020, the greatest grain yield was produced by
the 100% dose of NPK and co-inoculation with *R. tropici + A. brasilense + B. subtilis*, which
was statistically similar to the 100% dose of NPK and inoculation with *R. tropici, R. tropici
+ B. subtilis, R. tropici + P. fluorescens* and *R. tropici + A. brasilense + P. fluorescens*, as well
as the 50% dose of NPK and co-inoculation with *R. tropici + P. fluorescens* and *R. tropici +
A. brasilense + P. fluorescens* (Figure 4B). The lowest grain yield was observed following

the 0% dose of NPK fertilizer and inoculation with *R. tropici and R. tropici + A. brasilense + B. subtilis* (Figure 4B).

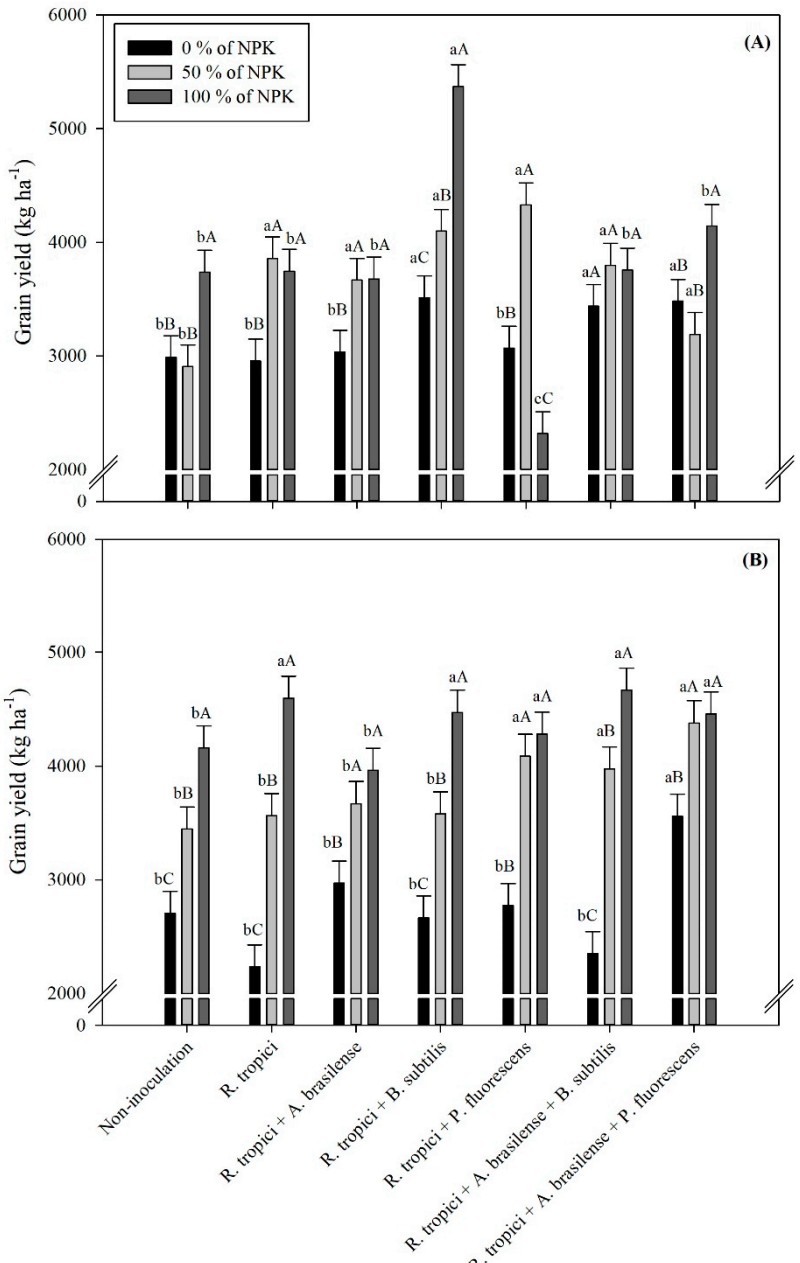

**Figure 4.** Effects of interactions between reduced doses of NPK fertilizer and co-inoculation with PGPB in common bean seeds on grain yield in 2019 (**A**) and 2020 (**B**). Means followed by the same capital letter differ from each other (different colors refer to the different doses of fertilizer with each inoculation), according to Tukey's test at a 5% probability. Means followed by different lowercase letters differ from each other (bars of the same color refer to each inoculation with different doses of NPK fertilizer), according to the Scott–Knott test at a 5% probability.

### 3.4. Correlations

We observed the following positive and significant correlations: hundred-grain weight, grain yield and the number of pods per plant with the number of grains per plant; the number of grains per pod with leaf P and K concentrations; the number of grains per plant with leaf P concentration; leaf N concentration with leaf P and K concentrations; leaf P concentration with leaf K concentration in 2019. In addition, we also observed the following

positive and significant correlations: hundred-grain weight, leaf N concentration and the number of pods per plant with the number of grains per pod$^{-1}$ and leaf K concentration; the number of grains per pod with the number of grains per plant and leaf N, P and K concentrations; grains per plant with grain yield and leaf N, P and K concentrations in 2020 (Figure 5).

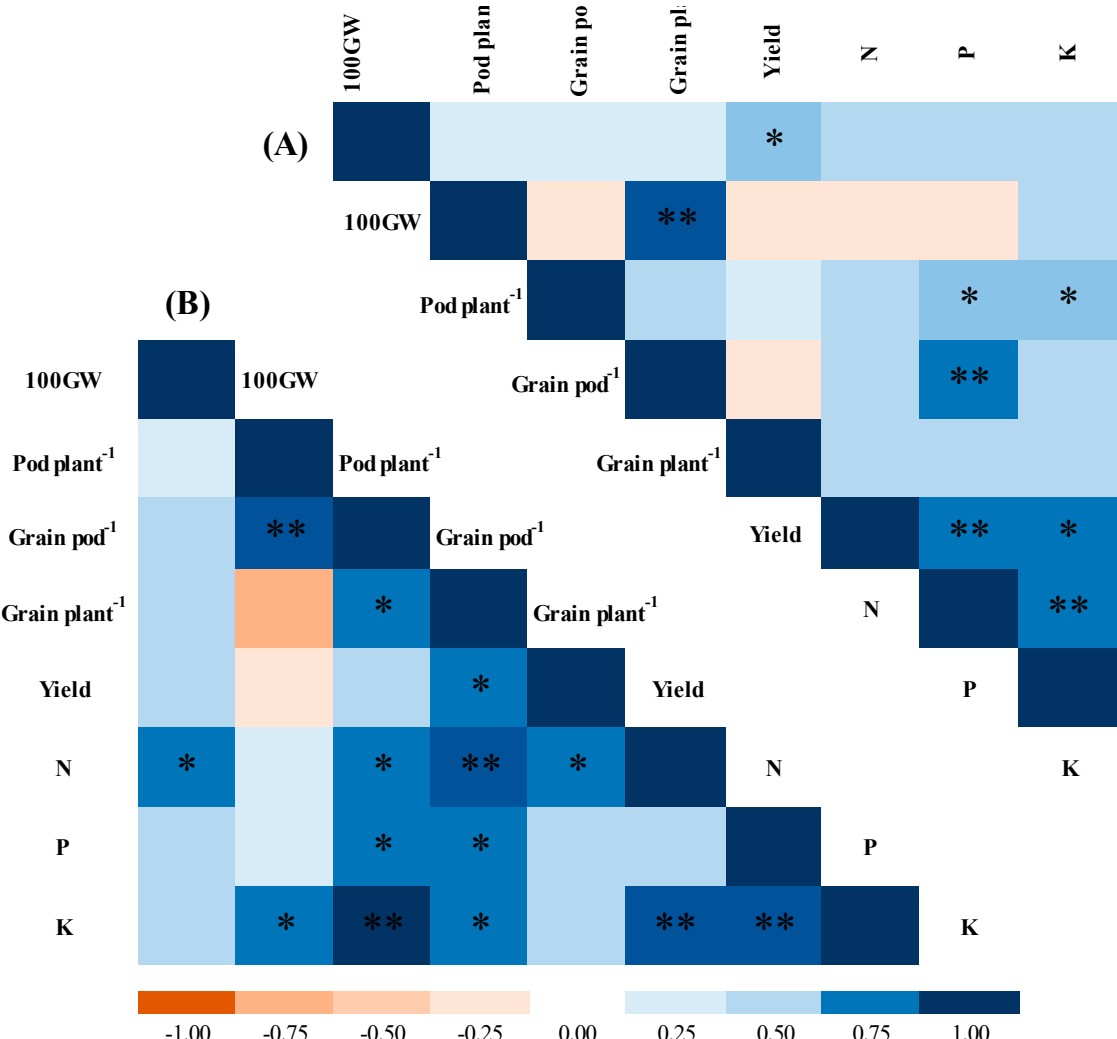

**Figure 5.** Heatmap of Pearson's correlation coefficients obtained from the analyzed variables of the common bean in response to NPK doses and co-inoculation with PGPB in common bean seeds in 2019 (**A**) and 2020 (**B**): ** significant correlation ($p < 0.01$); * significant correlation ($p < 0.05$); 100 GW, hundred-grain weight; YIELD, grain yield; N, leaf nitrogen content; P, leaf phosphorus content; K, leaf potassium content.

## 4. Discussion

The common bean leaf concentrations of N, P and K observed in the present experiments were within the adequate range or higher, as already described by Ambrosano et al. [17], i.e., between 30.0 and 50.0 g kg$^{-1}$ of N, between 2.5 and 4.0 g kg$^{-1}$ of P and between 20.0 and 24.0 g kg$^{-1}$ of K. The high nutritional status of the common bean plant during the flowering stage is due to the greater availability of nutrients during anthesis and grain filling, which consequently increases the hundred-grain weight, the number of pods, the number of grains per plant and grain yield [19,20].

The current results exhibited that the leaf N concentration of the common bean increased following the 100% dose of NPK fertilizer and co-inoculation with *R. tropici* + *A. brasilense* + *B. subtilis* and *R. tropici* + *A. brasilense* and the 0% dose of NPK and co-

inoculation with *R. tropici* + *A. brasilense* and *R. tropici* + *P. fluorescens* (Table 1; Figure 2A). This could be due to the greater expansion of the root system, which is stimulated by plant growth-promoting bacteria (PGPB). In addition, N has a great interference on the development and grain yield of bean plants because it is the most exported nutrient by plants; therefore, an increase in leaf N concentration is due to its availability in the rhizosphere region, which is affected by biological N fixation and the exploration capacity of the roots [21]. It has also been reported that different PGPB are able to colonize in root rhizospheres that have higher levels of nutrient transportation and concentration in the plants [22].

The dynamics of P in Brazilian soils are very complex and its absorption by plant roots does not depend on greater soil exploitation since, most of the time, P is adsorbed by the soils and is thus unavailable to plants. Therefore, the use of phosphate solubilizing bacteria, such as *P. fluorescens* and *B. subtilis*, could increase its availability in root rhizospheres and thus, plants could export it in greater quantities [14]. The present results indicated that the leaf P concentration increased following the 0 and 50% doses of NPK fertilizer and co-inoculation with *R. tropici* + *A. brasilense* + *P. fluorescens* and *R. tropici* + *B. subtilis* (Table 1; Figure 2B). Several studies have described how PGPB inoculation could adopt different mechanisms to enhance native populations and unlock inorganic P in a soluble form, which improves plant nutrition [11,23]. Phosphorus is also a part of the adenosine triphosphate (ATP) formation, which is the main source of energy for processes such as the transport of assimilates, photosynthesis and cell division [24] and is important for flower setting, grain filling and grain productivity [25].

The leaf K concentration of the common bean increased following the 50% dose of NPK fertilizer and co-inoculation with *R. tropici* + *A. brasilense* + *P. fluorescens* in 2019 and the 50% dose of NPK and inoculation with *R. tropici* in 2020 (Figure 2C,D). There was an increase of 17 and 19% in the leaf K concentration of the common bean in 2019 and 2020, respectively. This could be due to the ability of the bean to consume more K through efficient use, as it absorbed more than 50% of the total applied K at sowing [26]. Grover et al. [27] exhibited that increasing the K content in soil can increase its uptake by plants due to the increased soil exploitation of root growth that was stimulated as a result of phytohormone production in the rhizosphere region. Potassium regulates and participates in many important processes, such as photosynthesis, the opening and closing of stomata, water absorption from soil and the synthesis of ribulose bisphosphate carboxylase to improve leaf water status [28]. Beneficial microbes improve nutrient availability by exchange reactions and secreting organic acids that directly chelate silicon or dissolve rock potash to make them available for plant uptake, as well as contributing to the soil environment [29].

The hundred-grain weight increased following co-inoculation with *R. tropici* + *B. subtilis* and *R. tropici* + *P. fluorescens* and the 50% dose of NPK fertilizer (Table 2; Figure 3A). As reported earlier by Schossler et al. [30], the hundred-grain weight of the common bean increased more following co-inoculation with *R. tropici* + *A. brasilense* in relation to inoculation with *R. tropici*. The number of grains per pod increased with no inoculation and co-inoculation with *R. tropici* + *A. brasilense* and the 0% dose of NPK fertilizer (Table 2), while co-inoculation with *R. tropici* + *A. brasilense* + *P. fluorescens*, inoculation with *R. tropici* and no inoculation treatments with the 50% dose of NPK fertilizer were observed to produce higher numbers of grains per pod (Figure 3B). This could be due to the role of these bacteria in the transportation and uptake of nutrients and water by the roots to the plant shoot, providing greater gains and growth performance for the plants. It has also been reported that co-inoculation of the common bean with *R. tropici* + *P. fluorescens* + *A. lipoferum* increased the number of pods per plant and the number of grains per pod in relation to non-inoculation [31].

The highest number of grains per plant was produced by co-inoculation with *R. tropici* + *B. subtilis* and *R. tropici* + *A. brasilense* + *P. fluorescens* and the 0 and 50% doses of NPK fertilizer (Table 2; Figure 3C). A possible reason for this could be the role of *R. tropici* and *A. brasilense* in nitrogen acquisition by roots and the efficient biological nitrogen fixation

that leads to an increased number of pods and grains per plant in the common bean [10]. In addition, Jalal et al. [20] also described that the co-inoculation of *R. tropici* with *P. fluorescens* and *B. subtilis* produces a promising contribution to common bean production and nutrient acquisition in the tropical conditions of the Cerrado region in Brazil.

The grain yield of the common bean increased following the 100% dose of NPK fertilizer and co-inoculation with *R. tropici* + *B. subtilis* in 2019 and all co-inoculations, except *R. tropici* + *A. brasilense*, and non-inoculation in 2020. The co-inoculations with *R. tropici* + *P. fluorescens*, *R. tropici* + *A. brasilense* + *B. subtilis* and *R. tropici* + *A. brasilense* + *P. fluorescens* and the 50% dose of NPK fertilizer were observed to produce greater grain yields in both common bean harvests (Figure 4A,B). The main reason for the increased productivity could be the ability of *Azospirillum* sp. to colonize in the root rhizospheres of legumes with early nodulation and the production of flavonoids that attract the bacteria of *Rhizobium* sp. for further nodule formation [32,33]. Steiner et al. [10] demonstrated that the availability of readily soluble N from fertilizers is associated with the synergistic effects of bacteria and is favorable for plants with higher grain yields. Co-inoculation with *R. tropici* + *B. subtilis* and *R. tropici* + *P. fluorescens* increased the common bean yield due to the nutrient solubilization capacity of these bacteria, which makes nutrients available for absorption by the roots and transportation to the plant tissues [10]. The integrated use of fertilizers and PGPB is contributing to the N and organic matter contents in soils, which ultimately increase grain yields in a sustainable manner [34].

There were positive and significant correlations between the hundred-grain weight and grain yield in 2019 and between the grain yield in 2020 and the leaf N concentration and the number of grains per plant. The number of grains per plant and per pod had a positive and significant correlation with the leaf concentrations of N, P and K in 2020 while in 2019, the number of grains per pod had a positive and significant correlation with the leaf concentrations of P and K (Figure 5). The correlations observed in the present study also stated that due to the high accuracy of the data collected from the two harvests, there was a small environmental influence on the interactions between the PGPB and NPK fertilizer doses [35].

## 5. Conclusions

Co-inoculation with PGPB in association with NPK fertilizer application is one of the most sustainable strategies to improve the growth and yield of the common bean. This strategy, under the tropical conditions of Brazil, extended the grain yield by reducing plant dependency on synthetic fertilizers. Therefore, it was concluded that co-inoculation with *R. tropici* and *R. tropici* + *A. brasilense* and a 50% dose of NPK increased leaf N concentration in common bean, as did co-inoculation with *R. tropici* + *A. brasilense* and *R. tropici* + *P. fluorescens* and the 0% dose of NPK fertilizer. The leaf P concentration of the common bean was increased by inoculation with *R. tropici* and *R. tropici* + *B. subtilis* and the 50% dose of NPK fertilizer, as well as co-inoculation with *R. tropici* + *B. subtilis* and *R. tropici* + *A. brasilense* + *P. fluorescens* and a 0% dose of NPK fertilizer. Co-inoculation with *R. tropici* + *P. fluorescens* and the 50% dose of NPK, as well as all co-inoculations and the 0% dose of NPK fertilizer, increased the leaf K concentration of the common bean.

The grain yield and yield components of the winter common bean responded differently to co-inoculation and the application of NKP fertilizer. The hundred-grain weight and the number of grains per pod and per plant increased following co-inoculation with *R. tropici* + *P. fluorescens* and *R. tropici* + *A. brasilense* + *P. fluorescens* and a 50% dose of NPK fertilizer. Most of the inoculations with a 50% dose of NPK increased the winter grain yield of the common bean. The inclusion of PGPB in NPK mineral fertilizer for the winter cultivation of the common bean has a great potential to reduce the recommended dose to 50% due to higher use efficiency. Therefore, we recommend co-inoculation with *R. tropici* + *P. fluorescens* and *R. tropici* + *A. brasilense* + *P. fluorescens* and a reduced dose (50%) of NPK fertilizer for the winter cultivation of the common bean to increase macronutrient concentrations and grain yield.

**Author Contributions:** Conceptualization, E.S.M. and M.C.M.T.F.; methodology, E.S.M., A.J. and C.E.d.S.O.; software, C.E.d.S.O. and P.A.L.R.; validation, G.C.F., N.C.M.P. and A.J.; formal analysis, C.E.d.S.O. and P.A.L.R.; investigation, G.C.F., P.A.L.R. and A.J.; resources, M.C.M.T.F., E.S.M. and N.C.M.P.; writing—original draft preparation E.S.M. and A.J.; writing—review and editing, C.E.d.S.O., A.J., V.d.N. and M.E.d.S.; visualization, G.C.F. and N.C.M.P.; supervision, M.C.M.T.F., E.S.M. and A.J.; project administration, M.C.M.T.F.; funding acquisition, M.C.M.T.F. All authors have read and agreed to the published version of the manuscript.

**Funding:** This research received funding from Coordenação de Aperfeiçoamento de Pessoal de Nível Superior (CAPES), the World Academy of Science (TWAS) and Conselho Nacional de Desenvolvimento Científico e Tecnológico (CNPq), the second author's doctoral fellowship (CNPq/TWAS grant number: 166331/2018-0) and a productivity research grant (award number 311308/2020-1) for the corresponding author.

**Data Availability Statement:** Not applicable.

**Acknowledgments:** The authors would like to thank São Paulo State University (UNESP) and PROPG -UNESP for providing technical support and CAPES and CNPq for funding the study.

**Conflicts of Interest:** The authors declare no conflict of interest.

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
