# Peer review of "Co-Inoculations with Plant Growth-Promoting Bacteria in the Common Bean to Increase Efficiency of NPK Fertilization"

_agronomy, doi:10.3390/agronomy12061325_

Round 1

Reviewer 1 Report

Background information on common bean production in Brazil was provided by the authors, based on CONAD 2020/2021 publication on bean production in Brazil. However, the presentation of figures in the opening portions has to be clarified. For example the authors indicated that "Brazil is one of the main producer of common bean (Phaseolus vulgaris L.) with a 32 production of 3.136 thousand tons in an area of 2.923 thousand hectares, with an average 33 productivity of 1.079 kg ha−1" according to line 31-33. "3.136 thousand/ 2.923 thousand" should be simplified in a manner that it could be clearly understood.

Line 62- sp. after Pseudomonas should not be italised.

Line 193-196: "Means followed by same capital letter differ from each other in different colors (fer-193 tilization with NPK within each inoculation) by Tukey test at 5% probability, and means 194 followed by different lowercase letters differ from each other in bars of the same color 195 (inoculation within each fertilization with NPK) by Scott-Knott test at 5% probability" should form part of Figure 2 caption.

Author Response

We would like to express our gratitude to the reviewer who took the time to provide such a thorough review of our manuscript. We believe that his suggestion made our manuscript more direct and easier to follow. We have addressed all of the concerns raised and provided a point-by-point answer on how we handled each comment provided. Our answers will be in italic and highlighted in red color after each comment as well as highlighted in yellow color in the text. We hope that we have answered every inquiry to your satisfaction and also hope that you will find this version of publishable quality.

Thank and regards.

Comments to authors and responses

Background information on common bean production in Brazil was provided by the authors, based on CONAD 2020/2021 publication on bean production in Brazil. However, the presentation of figures in the opening portions has to be clarified. For example the authors indicated that "Brazil is one of the main producer of common bean (Phaseolus vulgaris L.) with a production of 3.136 thousand tons in an area of 2.923 thousand hectares, with an average productivity of 1.079 kg ha−1" according to line 31-33. "3.136 thousand/ 2.923 thousand" should be simplified in a manner that it could be clearly understood.

R: The authors are grateful for all your precious suggestions and well addressed all of your concerns in the main text.

Line 62- sp. after Pseudomonas should not be italised.

R: Yes. Thanks, it was addressed as suggested.

Line 193-196: "Means followed by same capital letter differ from each other in different colors (fer-193 tilization with NPK within each inoculation) by Tukey test at 5% probability, and means followed by different lowercase letters differ from each other in bars of the same color (inoculation within each fertilization with NPK) by Scott-Knott test at 5% probability" should form part of Figure 2 caption.

R: The authors appreciate your suggestion. We addressed your suggestion in the main text to regulate sentence flow. 

Reviewer 2 Report

Include soil Nitrogen content, pH, etc. taken before and after the study for both seasons (2019 & 2020), especially since the study is mainly about nitrogen efficiency. The values should be presented in table format with standard deviation or std. error next to it.

The plants’ response to inoculations, seven co-inoculations and the NPK fertilizer rate n

Trt. #

co-inoculations

NPK (%)

Leaf N concentration

(g kg−1)

Yield

Etc.

1

Un-inoculated control

0

50

100

2

R. tropici

0

50

100

3

R. tropici + A. brasilense

0

50

100

4

R. tropici + B subtilis

0

50

100

5

etc.

6

7

eed to be presented in a table format as shown below.

Author Response

We would like to express our gratitude to the reviewer who took time to provide such a thorough review of our manuscript. We believe that his suggestion made our manuscript more direct and easier to follow. We have addressed all of the concerns raised and provide a point by point answer on how we handled each comment provided. Our answers will be in italic and highlighted in red color after each comment as well as highlighted in yellow color in text. We hope that we have answered every inquiry to your satisfaction and also hope that you will find this version of publishable quality.

Thank and regards.

Comments to authors and responses:

Include soil Nitrogen content, pH, etc. taken before and after the study for both seasons (2019 & 2020), especially since the study is mainly about nitrogen efficiency. The values should be presented in table format with standard deviation or std. error next to it.

R: We are thankful to the reviewer. Here we do not agree with the reviewer. The main focus of the study was to evaluate the plant nutrition and productivity of the common beans in the Brazilian Cerrado region. In Brazil, because it is a tropical country and the dynamics of N in the soil are very complex, with several factors of N loss and relatively rapid mineralization of organic N, the researchers are not determining soil nitrogen content. The mineral N content in the soil can change dramatically from one day to the next, for example, there is a rainfall of 50 mm. That is, an analysis of this type has no correlation with what the plant can actually absorb from the soil.

The pH and other soil chemical attributes were already determined before the experiment and presented in the materials and methods section. This study is already including extensive evaluations regarding plant nutrition and productivity, which are the main focus of the study.

Moreover, this manuscript is a part of the master thesis of the first author, who didn’t have more available resources to do all soil analysis.

We hope, our respected reviewer will understand the situation and accept our answer.

The plants’ response to inoculations, seven co-inoculations and the NPK fertilizer rate need to be presented in a table format as shown below.

Sample table:

Trt. #

co-inoculations

NPK (%)

Leaf N concentration

(g kg−1)

Yield

Etc.

1

Un-inoculated control

0

50

100

2

R. tropici

0

50

100

3

R. tropici + A. brasilense

0

50

100

4

R. tropici + B subtilis

0

50

100

5

etc.

6

7

    R: We really appreciate reviewer suggestion. However, the experiment was carried out in randomized blocks with 4 replications in a 7 × 3 factorial scheme. The treatments were composed of inoculations and co-inoculations (1 = un-inoculated-control, 2 = R. tropici, 3 = R. tropici + A. brasilense, 4 = R. tropici + B. subtilis, 5 = R. tropici + P. fluorecens, 6 = R. tropici + A. brasilense + B. subtilis and 7 = R. tropici + A. brasilense + P. fluorecens) and reduced doses of NPK at sowing in coverage fertilization (0unfertilized, 50 and 100% of the recommended dose).

The tables of the current manuscript are correct that indicated isolated effect of the two factors, which were reported above. The reviewer's table suggestion is valid only when there is a significant interaction between the mentioned factors (PGPB inoculations or co-inoculations and doses of NPK). In case of significant interactions of inoculations and NPK doses for some evaluations, we choose and presented the results in the form of graphs. Therefore, it is suitable to keep the same table format as already presented in the tables.

We hope, the respected reviewer will agree with our table presentation after this explanation.

Thanks and sincere regards;

Authors
